# MXene/Cellulose Hydrogel Composites: Preparation and Adsorption Properties of Pb^2+^

**DOI:** 10.3390/polym16020189

**Published:** 2024-01-08

**Authors:** Qiang Yang, Jia Zhang, Hairong Yin, Junkang Guo, Shenghua Lv, Yaofeng Li

**Affiliations:** 1Shaanxi Key Laboratory of Comprehensive Utilization of Tailings Resources, College of Chemical Engineering and Modern Materials, Shangluo University, Shangluo 726000, China; 2School of Materials Science & Engineering, Shaanxi University of Science and Technology, Xi’an 710021, China; foshan83218593@163.com; 3College of Chemistry and Materials Science, Weinan Normal University, Weinan 714099, China; 15909250670@139.com; 4School of Environmental Science and Engineering, Shaanxi University of Science and Technology, Xi’an 710021, China; frankerry@163.com; 5College of Bioresources Chemical and Materials Engineering, Shaanxi University of Science and Technology, Xi’an 710021, China; lsh630623@yahoo.com

**Keywords:** cellulose hydrogel, MXene, adsorbent, adsorption isotherm

## Abstract

In this work, acrylic cellulose hydrogel, a typical natural polymer adsorbent, was modified using MXene through in situ polymerization to create a synthetic inorganic–polymer composite known as MXene/cellulose hydrogel. FTIR, XRD, SEM, and thermogravimetric analyses were applied to characterize the chemical structure and micromorphology. The MXene/cellulose hydrogel was utilized for the removal of Pb^2+^ from wastewater. Under optimal experimental conditions (initial Pb^2+^ concentration of 0.04 mol/L, adsorption time of 150 min, pH = 5.5, and MXene doping content of 50% at 30 °C), a maximum adsorption capacity of 410.57 mg/g was achieved. The MXene/cellulose hydrogel corresponded with the pseudo-second-order kinetic equation model and exhibited a better fit with the Freundlich isotherm model.

## 1. Introduction

Water pollution has become an imminent global environmental problem with the rapid development of urbanization, industry, and agriculture. Hexavalent chromium (Cr(VI)), a high-risk heavy metal, has adverse effects on aquatic ecosystems and human health. Common treatment methods for heavy metals in water include chemical precipitation, catalytic reduction, ion exchange, etc. [1,2,3]. In recent years, researchers have increasingly focused on adsorption treatments. Adsorption treatment not only addresses the challenge of separating heavy metal compounds that are easily soluble in water from the water body but also offers advantages such as simple operation and low environmental impact [4,5].

Cellulose hydrogel is a highly molecular polymer with a three-dimensional network structure formed by chemical cross-linking. Its molecular chain contains abundant hydroxyl, carboxyl, and amino functional groups, which exhibit strong chelating ability against heavy metal ions [6]. However, pure-cellulose hydrogels have issues such as weak mechanical strength and uneven structure that limit their applications [7]. To address this, a cellulose/poly(ethylene imine) composite hydrogel was prepared by grafting hyperbranched chain PEI onto a cellulose chain with an ECH crosslinking agent. The resulting hydrogel demonstrated excellent Cu^2+^ ion removal ability (285.7 mg/g), based on the principle that condensation and additional reactions occur between the polymer’s amine and ester parts with the crosslinker in the presence of ECH, ultimately forming a Schiff base structure [8]. Building upon this work, Feng et al. [9] synthesized a MXene/PEI-modified sodium alginate gel (MPA) for Cr(VI) removal. The active groups on PEI and the OH of MXene exhibited highly efficient adsorption properties for Cr(VI), with an adsorption capacity reaching 538.97 mg/g. It was found that MXene could effectively improve the mechanical strength of MPA gel. Therefore, the introduction of MXene into cellulose hydrogels not only improves the mechanical stability of the hydrogels, but also enhances their adsorption capacity.

MXene, a graphene-like two-dimensional material, has excellent chemical stability, selectivity, and hydrophilicity, making it a promising and efficient adsorbent [10,11]. The presence of functional groups (-F, -OH, and -O) and structural defects in MXene enhances its hydrophilicity, which is beneficial for the adsorption of various molecules in water such as heavy metal ions, organic dyes, and organic pollutants [12]. However, when used in water treatment applications, MXene faces the challenge of being easily dispersed and achieving solid–liquid separation in aqueous solutions. This may lead to secondary environmental effects. Therefore, improving its stability performance becomes necessary. For MXene materials specifically, mechanical stability is influenced by the existence of M-X bonds and surface functional groups [13]. By implementing appropriate functional modifications between MXene and other materials, the mechanical elasticity of composites based on MXene can be strengthened [14]. Hu et al. [15] reported a simple method for preparing an oxygen-containing MXene adsorbent using solutions of HgCl_2_ and Hg(NO_3_)_2_. Due to the large number of oxygen-containing functional groups on its surface, the adsorbent exhibits good adsorption towards Hg^2+^ in a mixed divalent cation metal solution. It has a high adsorption capacity for Hg^2+^, with close to 100% at pH = 5.0 for mercury-containing wastewater. Based on the above analysis, introducing MXene material into hydrogel matrices can improve the mechanical properties and the adsorption capacity.

In this work, we proposed a facile fabrication of MXene/cellulose hydrogel, which results in a synthetic inorganic–polymer composite where acrylic cellulose hydrogel, a typical natural polymer adsorbent, is modified by MXene through in situ polymerization. The structure and mechanical properties of the MXene/cellulose hydrogel are characterized. Subsequently, we study the adsorption performance of the MXene/cellulose hydrogel for Pb^2+^, including the kinetics of adsorption and the behavior of adsorption isotherms. Furthermore, the adsorption mechanism of the MXene/cellulose hydrogel is suggested.

## 2. Material and Method

### 2.1. Materials

Cellulose, ammonium persulfate (APS), acrylic acid (AA), acrylamide (AM), and N,N′-methylenebisacrylamide (MBA) were purchased from Aladdin Co., Ltd. (Shanghai, China). Lithium fluoride (LiF) was obtained from J&K Co., Ltd. (Qingdao, China). EDTA, chrome black T, hexamethylenetetramine, absolute ethanol (EtOH), and lead nitrate (Pb(NO_3_)_2_) were supplied by DaMao Chemical Co., Ltd. (Tianjing, China). Ti_3_AlC_2_ (99%, 400 mesh) was provided by BeigaerNano Materials Company (Ningbo, China).

### 2.2. Stripping Preparation of Single Layer MXene

First, 2.00 g of LiF (99%) powder was quickly added to a 40 mL HCl (6 mol/L) solution and dissolved under ultrasound for 15 min at room temperature. Then, 2.00 g of Ti_3_AlC_2_ (400 mesh) powder was added into the LiF transparent solution, stirred at 35 °C for 48 h, and diluted with a large amount of deionized water after the reaction. The supernatant was centrifuged at 3500 r/min until the pH of the supernatant was neutral, and the precipitate obtained via centrifugation was dispersed in a certain amount of deionized water, ultrasonically stripped at 25 °C for 1 h, and then centrifuged at 3500 r/min for 1 h. Finally, the precipitate was dried in a vacuum at 60 °C for 24 h.

### 2.3. The Preparation Process of MXene/Cellulose Hydrogel Composite

Figure 1 showed schematic of the preparation concept of MXene/cellulose hydrogel. First, 2.0 g cellulose was dispersed in 30 mL deionized water, and then 2.0 g APS and a certain mass fraction (0%, 10%, 30%, 50%, 70%) of MXene solution were added. The mixture was ultrasonically dispersed for 20 min at room temperature. Next, 10 mL AA, 2.0 g AM, and 0.06 g MBA were added in sequence, stirred uniformly, and reacted to 50 °C for 8 h. After the reaction was completed, they were washed with deionized hydrated EtOH to neutrality [16].

### 2.4. Characterization

A VERTEX-80 (BRUKER, Mannheim, Germany) Fourier infrared spectrometer (FTIR) was used for structural characterization, with KBr pressed tablets, and the test range was 600~4000 cm^−1^. The phase analysis was carried out using a X’Pert Powder type (PANalytical, Almelo, The Netherlands) X-ray diffractometer (XRD) with acceleration voltage of 40 kV, scanning speed of 0.2°/S, CuKα, and λ = 0.1546 nm. The micro-morphology was observed using an S-4800 (HITACHI, Tokyo, Japan) scanning electron microscope (SEM), with a scanning voltage of 5 kV and Pt treatment. The thermal stability of the samples was characterized using a Dicovery 650 (BRUKER, Billerica, MA, USA) Synchronous Thermal Comprehensive Analyzer (SDT), and the test conditions were N_2_ atmosphere, 60 mL/min flow rate, and the measurement temperature was from room temperature to 600 °C at 10 °C/min. X-ray photoelectronspectroscopy (XPS) (S-4800, FEI, Tokyo, Japan) was used to explain the mechanism of Pb^2+^ adsorption by the MXene/cellulose hydrogel composite.

### 2.5. Adsorption Performance

We accurately pipetted 150 mL 0.01 mol/L Pb(NO_3_)_2_ solution into a 250 mL Erlenmeyer flask, then added HAc-NaAc buffer solution and hexamethylenetetramine solution dropwise to adjust the pH value. Afterward, 0.50 g of composite hydrogel material was added, and the adsorption experiment was carried out in a constant temperature oscillator (120 r/min). After reaching the adsorption equilibrium, the supernatant liquid was absorbed for determination. The adsorption *Q_e_* (mg/g) was calculated according to Equation (1):(1)$$Q_{e}=\frac{v(c_{0}-c_{e})}{m}$$
where *Q_e_* was the adsorption capacity of the adsorbent for Pb^2+^, mg/g; *c*_0_ was the initial concentration of the Pb^2+^ solution, mg/L; *c_e_* was the concentration of the Pb^2+^ solution after adsorption equilibrium, mg/L; *v* was the volume of the Pb^2+^ solution, L; and *m* was the amount of MXene/cellulose hydrogel, g.

## 3. Results and Discussion

### 3.1. Structure and Morphology Analysis

#### 3.1.1. FT-IR Analysis

The FTIR spectra of MXene/cellulose hydrogels with different doping contents (0%, 10%, 30%, 50%, and 70%) are shown in Figure 2. The absorption peaks at 1160 cm^−1^ and 1452 cm^−1^ are the characteristic absorption peaks of cellulose. After AA and AM were cross-linked, the -OH characteristic peak originally belonging to cellulose disappeared [17]. The stretching vibration absorption peaks of CO-NH and the asymmetric stretching vibration absorption peaks of -COOH appear at 1525 cm^−1^ and 1631 cm^−1^, respectively, while the characteristic absorption peaks of N-H appeared at 3462 cm^−1^. After treating Ti_3_AlC_2_ with the HCl-LiF mixed system, the broad absorption peak at 3458 cm^−1^ corresponds to the stretching vibration peak of the free hydroxyl group -OH [18], indicating successful stripping of Ti_3_AlC_2_ and synthesis of Ti_3_C_2_T_x_ with -OH on its surface. The analysis of the above results indicates that the MXene-doped cellulose hydrogel composite material was successfully prepared, and the surface of the MXene/cellulose hydrogel was rich in active adsorption sites.

#### 3.1.2. XRD Analysis

The XRD curves of MXene/cellulose hydrogels with different doping levels are demonstrated in Figure 3. It is observed that the characteristic diffraction peak at 2θ = 22.3° belonged to the cellulose hydrogel’s (002) crystal plane [19]. From the curve of MXene/cellulose hydrogel doped with 0%, it can be seen that the diffraction peak at 2θ = 42.0° is attributed to Al in Ti_3_AlC_2_ [20]. After a peeling treatment using the LiF/HCl system, new diffraction peaks appeared at 2θ = 9°, 18.2°, and 26.8°, indicating successful removal of Al and the formation of a new MAX phase, confirming the synthesis of a layered MXene material [21]. Combined with the data analysis results of Figure 2, it is evident that MXene/cellulose hydrogel composites with different doping amounts were successfully prepared.

#### 3.1.3. SEM Morphology

Figure 4 is SEM images of the MXene/cellulose hydrogel. As shown in Figure 4a, the pure cellulose hydrogel exhibits a smooth sheet-like structure, indicating a state of stacking layers. The preparation of cellulose hydrogel generally involves uniformly disperse cellulose in the reaction system before the polymer cross-linking reaction takes place. When AA and AM undergo polymerization to form polyacrylamide molecules, the composite material is induced to produce a sheet structure. The images of MXene/cellulose hydrogel composites with different doping contents are shown in Figure 4b–e. After being modified by MXene, the sheet-like cellulose hydrogel transformed into a non-smooth cluster structure, indicating that the nano-scale sheet structure of MXene would also grow inside the cellulose hydrogel in an interleaved manner due to its combination with hydrogel surface active sites. The infrared spectrum revealed the presence of an O-H absorption peak in MXene, suggesting that the MXene/cellulose hydrogel would undergo a coordination complex reaction against heavy metal ions to increase the adsorption capacity [22]. Additionally, the content of MXene on the cellulose gel surface initially increased first and then decreased. When the modified amount of MXene was 30%, the MXene deposition effect on the cellulose gel surface was the best, but the agglomeration effect was obvious. However, when the modified amount of MXene was ≥50%, the deposition amount of MXene gradually decreased, because when the content of MXene was low, hydrogen bonds could be formed between its reactive -OH, which could effectively improve the interaction with cellulose hydrogels. As the MXene content increased, steric hindrance between MXene–MXene also increased, reducing the surface activity of the composite and resulting in partial MXene.

#### 3.1.4. Thermogravimetric Analysis

The TG/DTG curves of the MXene/cellulose hydrogel composite under N_2_ atmosphere are shown in Figure 5a,b. The thermal weight loss process of the composite material can be divided into three stages: The first stage (RT~190 °C) mainly includes the removal of bound water within the hydrogel and water adsorbed on the surface of the MXene and HF molecules at high temperatures. In the second stage (190 °C~500 °C), organic solvents and cellulose undergo thermal cracking, resulting in carbon dioxide and other non-flammable gases such as carbon oxide chemicals. As displayed in Figure 5c, the pyrolysis of organic solvent and cellulose occurs at 260 °C in the MXene/cellulose hydrogel composite. This is due to the violent movement of molecular chains at high temperature, effectively overcoming the hydrogen bonds and breaking down the internal structure of cellulose hydrogel, thereby reducing sample stability [23]. In the third stage (500 °C~800 °C), chemical bonds between the active sites are destroyed at high temperature, leading to complete thermal decomposition of the polyacrylamide polymer. Due to an endothermic effect, a sample strong melting peak was observed at 500 °C.

In addition, in relation to Figure 5a,b, the residual amounts of sample with doping contents of 10%, 50%, and 70% show a linear increase trend for MXene. At 630 °C, the residual content of the 70% was measured as 12.5%, while the sample doped with 50% had a residual content of 11.0%. However, in the second stage, the composite with a doping content of 50% exhibited preferential thermal stability. Therefore, on the whole, the thermal stability of the MXene/cellulose hydrogel composite doped with 50% was found to be superior.

### 3.2. The Adsorption Performance of MXene/Cellulose Hydrogel for Pb^2+^

#### 3.2.1. Effect of Adsorption Time

The initial concentration of the Pb^2+^ solution was 0.01 mol/L, the pH was 5.5, and the adsorption temperature was 30 °C. The effect of adsorption time for the adsorption capacity of MXene/cellulose hydrogel is shown in Figure 6a. As shown in Figure 6a, the adsorption process of Pb^2+^ by composite aerogel can be divided into three stages. In the initial stage of adsorption (*t* ≤ 60 min), Pb^2+^ experiences chemical electrostatic attraction with active groups (-COOH, -NH_2_, -OH) present in the gel composite material, leading to a rapid increase in its adsorption capacity. During the middle stage of adsorption (60 min < *t* ≤ 150 min), saturation occurs gradually at the surface sites of the composite hydrogel resulting in a slower rate for Pb^2+^ uptake. Pore diffusion becomes dominant during this stage as a mechanism for Pb^2+^ absorption. In the final stage of adsorption (*t* > 150 min), all available active sites become occupied and further absorption ceases, reaching an equilibrium state essentially. Consequently, due to electrostatic attraction forces, Pb^2+^ readily combines with the active groups (-COOH, -NH_2_, -OH) on the surface of the hydrogel composite material, which enhances its overall efficiency within first 60 min. On the other hand, the dispersed MXene sheet within gel structure exhibits slow but continuous absorption towards Pb^2+^, requiring longer duration until reaching equilibrium [24].

#### 3.2.2. Effect of pH

The initial concentration of the Pb^2+^ solution was 0.01 mol/L, the adsorption time was 150 min, and the adsorption temperature was 30 °C. The effect of the pH value on the adsorption capacity of the MXene/cellulose hydrogel composite material for Pb^2+^ ions was investigated, as shown in Figure 6b. With the increased in pH, the adsorption capacity of MXene/cellulose hydrogel composite shows a linear increase trend. When the pH value was 5.5, the adsorption capacity of the MXene/cellulose hydrogel composite material to Pb^2+^ ions reached the maximum, which was 200.14 mg/g. At lower pH values, there would was a large amount of H^+^ in the solution, and H^+^ and Pb^2+^ competed for adsorption sites. At the same time, the ionization process of -COOH and -OH was inhibited from a more acidic solution, which is not conducive to the formation of complexes, so the adsorption capacity was small [25]. When the pH value gradually increased, the content of H^+^ in the solution gradually decreased, making the adsorption sites relatively increased. Thus, -COOH was easily ionized into -COO-, and -COO- formed a complex about Pb^2+^ to increase the adsorption capacity. Sharma et al. [26] reported that excessively high H+ concentrations above pH 5.5 led to greater hydrolysis of Pb2+, forming complexes that were difficult to absorb and ultimately reducing overall absorption.

#### 3.2.3. Effect of Adsorption Temperature

The initial concentration of the Pb^2+^ solution was selected to be 0.01 mol/L, the adsorption time was 150 min, and the pH value was 5.5. The relationship between the adsorption temperature and the adsorption capacity of MXene/cellulose hydrogel is shown in Figure 6c. It can be seen that with an increase in temperature, the adsorption amount of Pb^2+^ by the hydrogel composite material initially increases and then decreases. The 50% doped MXene/cellulose hydrogel exhibits its highest adsorption capacity at 30 °C, which is measured a 205.26 mg/g. An increase in temperature promotes molecular movement speed, allowing more contact between the hydrogel and Pb^2+^, thereby increasing reaction probability and improving adsorption efficiency [27]. However, when the adsorption temperature exceeds 30 °C, there is a gradual decrease in the adsorption capacity of MXene/cellulose hydrogel due to exothermic reactions between -COOH groups and Pb^2+^ [28]. Additionally, a high-temperature environment accelerates thermal movement of metal ions in the solution leading to increased collisions among them, ultimately reducing overall adsorption capacity.

#### 3.2.4. Effect of the Initial Concentration of Pb^2+^

The pH value of the Pb^2+^ solution was set at 5.5, the adsorption time was 150 min, and the adsorption temperature was 30 °C. The effect of the initial concentration of the Pb^2+^ solution on the adsorption capacity of the MXene/cellulose hydrogel is shown in Figure 6d. It demonstrates that as the concentration of Pb^2+^ increased, the amount of Pb^2+^ adsorbed by the MXene/cellulose hydrogel also increased. When the initial concentration of Pb^2+^ was 0.04 mol/L, the maximum adsorption capacity of Pb^2+^ by MXene/cellulose hydrogel reached 410.57 mg/g. With an increase in metal ion concentration, there is a greater driving force for mass transfer and a reduction in adsorption resistance, which leads to an increased probability of contact between Pb^2+^ and adsorption sites and helps improve the adsorption performance of composite materials [29]. However, when the concentration exceeds 0.04 mol/L, it can lead to increased hydrolysis of Pb^2+^, reducing its tendency to bind with active groups and resulting in a decrease in the composite material’s adsorption capacity [30].

Based on the analysis results from Figure 6, the optimal adsorption conditions for MXene/cellulose hydrogels were determined as follows: Pb^2+^ concentration of 0.04 mol/L, an adsorption time of 150 min, adsorption temperature of 30 °C, and pH = 5.5. Under these conditions, the MXene/cellulose hydrogel with a doping level of 50% exhibits the best adsorption effect, with a maximum adsorption capacity of 410.57 mg/g.

### 3.3. Adsorption Isotherm Behavior

The process of the removal of Pb^2+^ by 50% doped MXene/cellulose hydrogel was carried out using the Langmuir and Freundlich isotherm models of equilibrium modeling:(2)$$\mathrm{Langmuir}:\frac{C_{e}}{Q_{e}}=\frac{C_{e}}{Q_{m}}+\frac{1}{k_{1}Q_{m}}$$
(3)$$\mathrm{Freundlich}:\mathrm{lg}Q_{e}=\mathrm{lg}k_{f}+\mathrm{lg}\frac{C_{e}}{n}$$
where *C_e_* is the Pb^2+^ concentration after adsorption equilibrium, mg/L; *Q_m_* is the saturated adsorption capacity, mg/g; *k*_1_ is the adsorption equilibrium constant; *k_f_* is the Freundlich adsorption constant; and *n* is the Freundlich heterogeneity factor.

The obtained constants and parameters for Langmuir and Freundlich isotherms are listed in Table 1, and the fitting curves of MXene/cellulose hydrogel for Pb^2+^ adsorption using each isotherm model are shown in Figure 7. The correlation coefficients of Langmuir and Freundlich are 0.51702 and 0.97093, respectively. In general, when the Freundlich constant 1/*n* is between 0.1 and 0.5, adsorption is likely to occur. When it is greater than 2, adsorption is difficult to occur. Thus, Freundlich isotherm model is more suitable to describe the adsorption process of Pb^2+^ onto MXene/cellulose hydrogel. This indicates that the adsorption process of MXene/cellulose hydrogel on Pb^2+^ results from combined effect sites and MXene.

### 3.4. Kinetics of Adsorption

The process of the removal of Pb^2+^ by 50% doped MXene/cellulose hydrogel was evaluated using pseudo-first-order and pseudo-second-order kinetic models. In the adsorption experiment, kinetic parameters are important for predicting the adsorption rate and equilibrium time, and can better explain the role of the adsorbent in the adsorption process. The calculation formulas for the pseudo-first-order and pseudo-second-order kinetic model can be explained as follows [31]:(4)$$\text{pseudo-first-order}\;\mathrm{model}:\ln(Q_{e}-Q_{t})=\ln Q_{e}-\frac{k_{1}}{2.303}t$$
(5)$$\text{pseudo-second-order}\;\mathrm{model}:\frac{t}{Q_{t}}=\frac{1}{k_{2}Q_{e}^{2}}+\frac{t}{Q_{e}}$$
where *Q_e_* and *Q_t_* are the amounts of Pb^2+^ ion adsorbed at equilibrium and at time *t* (min), mg/g, respectively. *k*_1_ (min^−1^) and *k*_2_ (g/mg·min) are the rate constants of the pseudo-first-order adsorption and pseudo-second-order adsorption. The kinetic model was further used to analyze the adsorption process and to discuss the possible mass transfer power and diffusion control in the adsorption process.

The kinetic model was further used to analyze the adsorption process and discuss the possible mass transfer power and diffusion control in it. The values of *k*_1_, *k*_2_, and *Q_e_* along with those of the correlation coefficient (R^2^) are summarized in Table 2. The fitting curves of both the pseudo-first-order and pseudo-second-order kinetic model are shown in Figure 8. In the pseudo-second-order kinetics, the calculated *Q_e_* is nearly identical to experimental values, and the R^2^ values were found to be 0.9784. This indicates that the pseudo-second-order model is appropriate for describing the adsorption phenomena.

### 3.5. XPS

The XPS spectra of MXene/cellulose hydrogel before and after Pb^2+^ absorption are shown in Figure 9a. It is evident that successful adsorption of Pb^2+^ occurred when the photoelectron spectral line of the Pb 4f element appears in 148 eV following Pb^2+^ adsorption by the material. The N1s XPS spectra before and after Pb^2+^ adsorption of MXene/cellulose hydrogel are displayed in Figure 9b and Figure 9c, respectively. Prior to Pb^2+^ adsorption, photoelectron spectral lines of nitrogen atoms in primary and secondary amine groups appear at 399.39 eV and 401.36 eV. The photoelectron spectra of nitrogen atoms in primary and secondary amine groups were transferred to 399.47 eV and 401.47 eV following the adsorption of Pb^2+^, respectively. Both primary and secondary amines were verified to be involved in the chelation of Pb^2+^. The O1s XPS spectra before and after Pb^2+^ adsorption of MXene/cellulose hydrogel are displayed in Figure 9d,e, respectively. It was discovered that the peaks at 530.62 eV and 529.62 eV shifted (to 530.75 eV and 529.75 eV, respectively). A coordination bond was formed in the material by the O atom and Pb^2+^ component.

### 3.6. Suggested Adsorption Mechanism

In summary, the adsorption process of MXene/cellulose hydrogel composites follows a pseudo-second-order kinetic equation model and is better described by the Freundlich model. The maximum absorption capacity of the obtained MXene/cellulose hydrogel reached 410.57 mg/g. It can be inferred that physical adsorption dominates during the initial stage of adsorption for MXene/cellulose hydrogel, where intermolecular attraction between active sites in the composite gel played a major role in Pb^2+^ uptake. In later stages, chemical adsorption becomes predominant. As depicted in Figure 10, coordination occurred between -COOH, -NH_2_, -CO-NH_2_ groups on the composite gel with Pb^2+^ [32]. Additionally, a substantial amount of O-H groups on MXene nanosheets underwent an ion exchange reaction with Pb^2+^ [33].

## 4. Conclusions

In this study, a modified cellulose hydrogel incorporating MXene (Ti_3_AlC_2_), which is a novel inorganic–organic composite absorbent, was demonstrated to be an effective adsorbent for the removal of Pb^2+^ ions from wastewater.

(1)The optimized adsorption conditions for MXene/cellulose hydrogel were Pb^2+^ initial concentration 0.04 mol/L, 150 min, 30 °C, pH = 5.5, and MXene doping content of 50%. Its adsorption capacity attained 410.57 mg/g.(2)The analysis of adsorption isotherm and kinetics described that the adsorption process of MXene/cellulose hydrogel follows the pseudo-second-order kinetic equation model and is better fitted with the Freundlich model. The experimental data suggest that the MXene/cellulose hydrogel composite has good adsorption capacity for Pb^2+^ under high concentration and acidic conditions, making it a suitable absorbent material in the field of wastewater treatment of heavy metal ions.

## Figures and Tables

**Figure 1 polymers-16-00189-f001:**
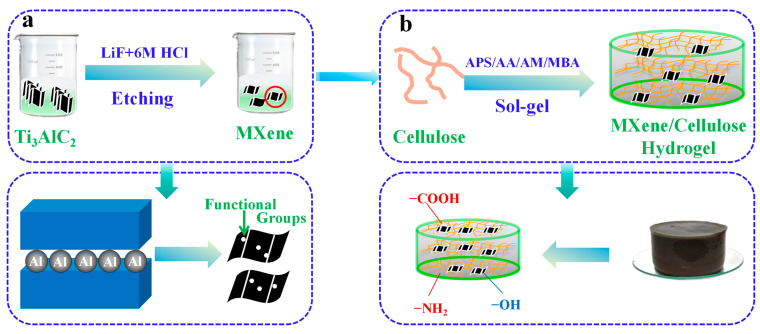
Schematic of the preparation concept of MXene/cellulose hydrogel ((**a**) the preparation process of single layer MXene; (**b**) the preparation process of MXene/Cellulose hydrogel composite).

**Figure 2 polymers-16-00189-f002:**
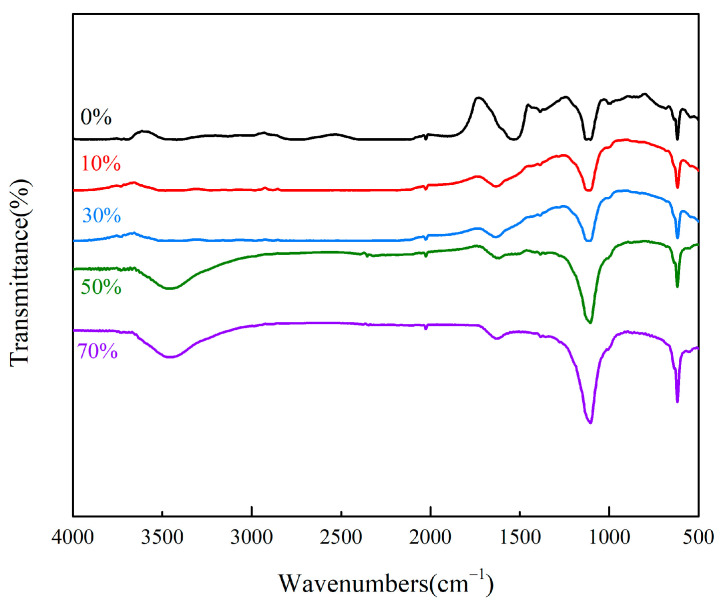
FTIR spectra of MXene/cellulose hydrogels with different doping mass.

**Figure 3 polymers-16-00189-f003:**
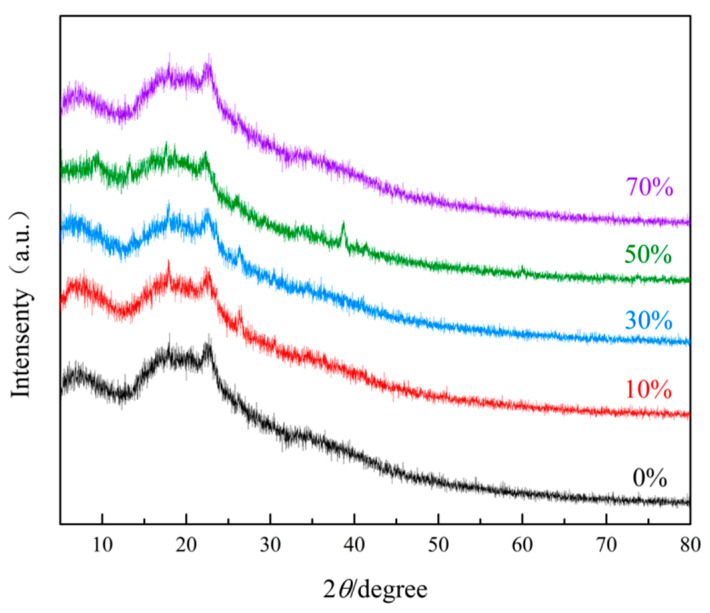
XRD curves of MXene/cellulose hydrogels with different doping mass.

**Figure 4 polymers-16-00189-f004:**
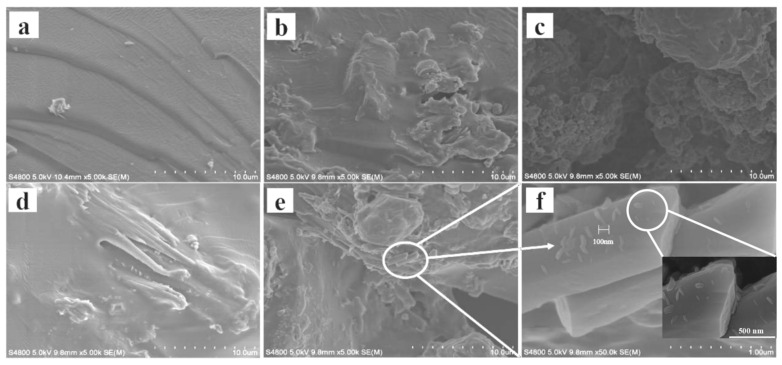
The SEM images of MXene/cellulose hydrogel composites ((**a**) 0%; (**b**) 10%; (**c**) 30%; (**d**) 50%; (**e**) 70%; (**f**) 70% MXene/cellulose hydrogel enlarged view).

**Figure 5 polymers-16-00189-f005:**
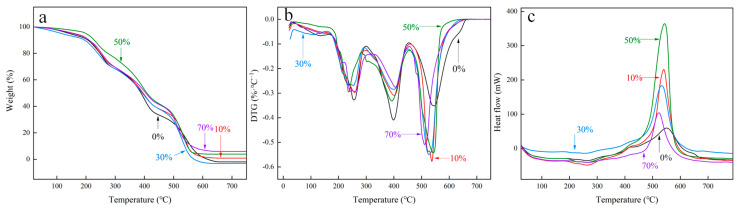
The TG-DTA-DSC curves of MXene/cellulose hydrogel ((**a**) TG; (**b**) DTG; (**c**) DSC).

**Figure 6 polymers-16-00189-f006:**
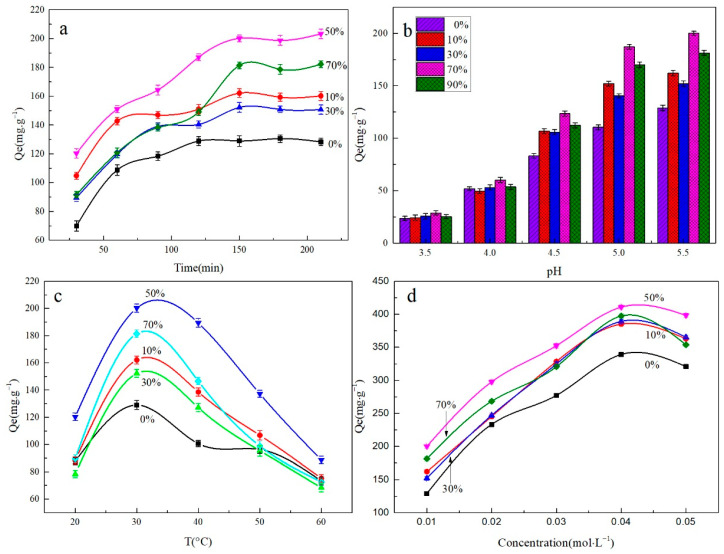
The adsorption performance of MXene/cellulose hydrogel ((**a**) effect of adsorption time on the removal of Pb^2+^; (**b**) effect of pH value on the removal of Pb^2+^; (**c**) effect of adsorption temperature on the removal of Pb^2+^; (**d**) effect of initial Pb^2+^ concentration on the removal of Pb^2+^).

**Figure 7 polymers-16-00189-f007:**
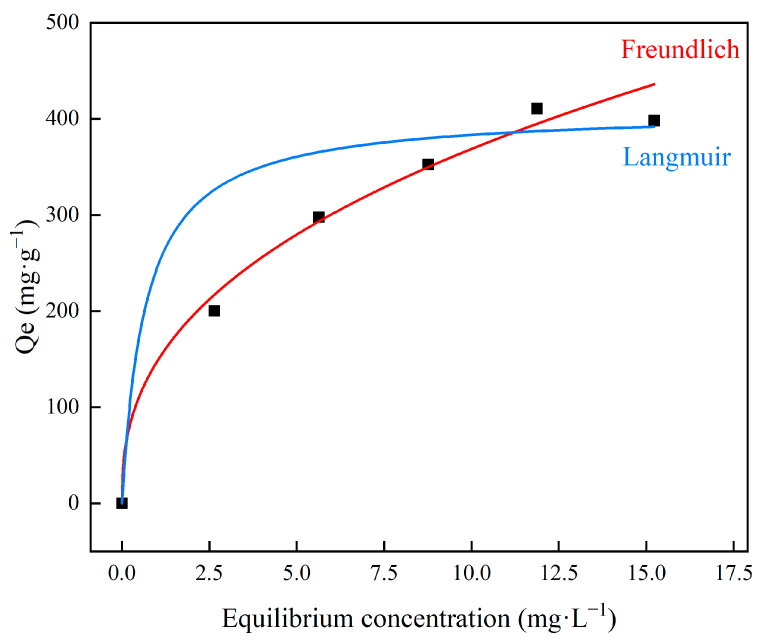
The Langmuir and Freundlich isotherms for removal of Pb^2+^ by MXene/cellulose.

**Figure 8 polymers-16-00189-f008:**
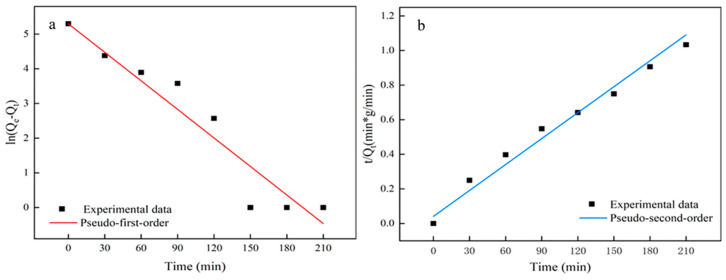
Kinetic model fitting curves of MXene/cellulose hydrogel ((**a**) pseudo-first-order; (**b**) pseudo-second-order).

**Figure 9 polymers-16-00189-f009:**
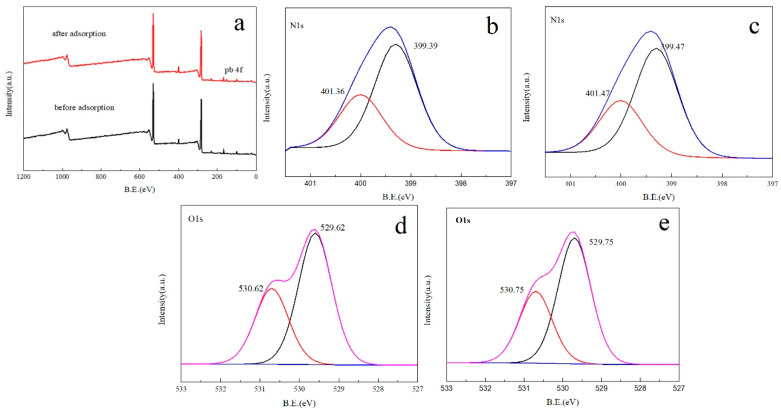
XPS spectrum of MXene/cellulose hydrogels composites (**a**) and high-resolution scan of N1s (**b**,**c**), and O1s (**d**,**e**) before and after Pb^2+^.

**Figure 10 polymers-16-00189-f010:**
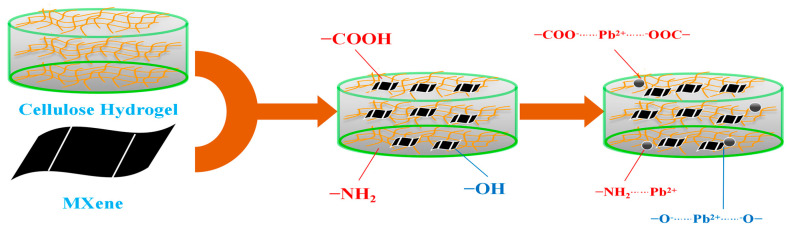
The suggested adsorption mechanism.

**Table 1 polymers-16-00189-t001:** Parameters of Langmuir and Freundlich isotherms.

Isotherms	Langmuir			Freundlich		
Parameters	*Q_m_*	*k* _l_	R^2^	*n*	*k_f_*	R^2^
	412	1.48598	0.51702	2.50794	147.2	0.97093

**Table 2 polymers-16-00189-t002:** Pseudo-first-order and pseudo-second-order kinetic parameters.

	Pseudo-First-Order			Pseudo-Second-Order		
Parameters	*k*_1_ (min^−1^)	Q_e_ (mg/g)	R^2^	*k*_2_ (g/mg·min)	Q_e_ (mg/g)	R^2^
	0.0274	200.13	0.9189	0.6146	410.57	0.9784

## Data Availability

The data presented in this study are available on request from the corresponding author.

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
