# Peer review of "MXene/Cellulose Hydrogel Composites: Preparation and Adsorption Properties of Pb2+"

_polymers, 2024, doi:10.3390/polym16020189_

Round 1

Reviewer 1 Report

Comments and Suggestions for Authors

The authors reported the fabrication of a modified MXene/cellulose hydrogel composite, which had the ability to remove Pb2+ in wastewater. Detailed characterizations of chemical structure and morphology have been investigated. The optimal conditions for the hydrogel composite to adsorb Pb2+ and the potential adsorption mechanism have also been studied. While the data presented in this manuscript is adequate, there are several major comments that the authors may want to consider:

1) In the introduction, the transition between cellulose hydrogels to MXene/cellulose hydrogels needs to be improved. Why do the improved conductivity and mechanical property matter with the adsorption ability of the hydrogel composite? Is it true that adding MXene makes the cellulose hydrogel more biocompatible?

2) There are too many small figures. Please consider combining them into 4-6 big figures. The axes of Figures 5, 10, and 11 are in Chinese, please translate them into English. Figures 10 and 11 are in the wrong order.

3) Figures 8 and 9 are exactly the same. The actual Figure 9 is missing for Section 3.4.5. 

4) The XRD curve in Figure 2 for 0% and 70% look exactly the same except for the color. Is the data plausible?

5) There are no standard deviations in all the experimental data and figures. Is the data repeatable and representative?

6) The mechanical testing and analysis do not make sense. First, the authors should not use their hands to compress or stretch the samples to test the mechanical properties of the samples. Second, the other compositions for 30%, 50%, and 70% are missing.

Overall, I believe a major revision is needed to polish this work before the next submission.

Comments on the Quality of English Language

Although it doesn't affect the overall understanding, there are many grammar mistakes and misspellings. Please proofread the manuscript carefully.

Reviewer 2 Report

Comments and Suggestions for Authors

This paper is more like a lab work report instead of an academic paper. The authors didn’t provide a sound explanation of some experiments (like 3.2, and 3.3). They are more like the stack of experimental data along with the corresponding description. In addition, Figure 8 and Figure 9 look like exactly the same picture, I’m questioning the authors' honesty. Figure 10 and Figure 11 have foreign characters in the figures. Please double-check all figures’ format before submitting your paper. It’s very unprofessional. 

Comments on the Quality of English Language

Many sentences need to be corrected. 

Round 2

Reviewer 1 Report

Comments and Suggestions for Authors

I recommend accepting the modified manuscript after a minor grammar check.

Reviewer 2 Report

Comments and Suggestions for Authors

Good for publishing.

Comments on the Quality of English Language

None.
